# Rethinking Gradient Weight’s Influence over Saliency Map Estimation

**DOI:** 10.3390/s22176516

**Published:** 2022-08-29

**Authors:** Masud An Nur Islam Fahim, Nazmus Saqib, Shafkat Khan Siam, Ho Yub Jung

**Affiliations:** Department of Computer Engineering, Chosun University, Gwangju 61452, Korea

**Keywords:** class activation map (CAM), explainable AI (XAI), global guidance map

## Abstract

Class activation map (CAM) helps to formulate saliency maps that aid in interpreting the deep neural network’s prediction. Gradient-based methods are generally faster than other branches of vision interpretability and independent of human guidance. The performance of CAM-like studies depends on the governing model’s layer response and the influences of the gradients. Typical gradient-oriented CAM studies rely on weighted aggregation for saliency map estimation by projecting the gradient maps into single-weight values, which may lead to an over-generalized saliency map. To address this issue, we use a global guidance map to rectify the weighted aggregation operation during saliency estimation, where resultant interpretations are comparatively cleaner and instance-specific. We obtain the global guidance map by performing elementwise multiplication between the feature maps and their corresponding gradient maps. To validate our study, we compare the proposed study with nine different saliency visualizers. In addition, we use seven commonly used evaluation metrics for quantitative comparison. The proposed scheme achieves significant improvement over the test images from the ImageNet, MS-COCO 14, and PASCAL VOC 2012 datasets.

## 1. Introduction

Deep neural networks have achieved superior performance on numerous vision tasks [1,2,3]. However, they contain complicated black boxes with huge, unexplainable parameters that begin with random initialization and reach another unpredictable (still) sub-optimal point. Such transformations are non-linear for each problem, the interpretability remains unsolved.

The vision community relies on estimating saliency maps for deciphering the decision-making process of deep networks. We try to bridge the unknown gap between the input space and decision space through this saliency map. These saliency map generation approaches can be divided into different procedures regarding input space [4,5,6], feature maps [7,8,9,10,11,12,13], or propagation scheme [14,15,16,17,18]. Usual perturbation-based methods [19,20,21] probe the input space into different deviated versions and obtain a unified saliency map by their underlying algorithms. Even though their approach produces a better result, their process is relatively slow. CAM-like methods rely on the gradient for the given class and create a decent output within a very brief computation time.

This study explores the established framework of CAM-based methods to address current issues within vision-based interpretability. Similar studies usually rely on gradient information to produce saliency maps. Some replace the gradient dependency with score estimation to build the saliency map out of the feature maps. Nonetheless, these studies formulate saliency maps that often suffer degradation during class discriminative examples. Moreover, their weighted accumulation does not always cover the expected region for the given single class instance, as the associated weights sometimes do not address the local correspondence. On the other hand, the gradient-less methods take significant time to produce a score index for the feature maps.

Typically, the gradient maps are averaged into single values to produce the saliency map within CAM-like studies. However, this projection is not always efficient for the saliency map. In Figure 1, we can see that weighted multiplication still contains traces of unwanted classes. For dual-class image (right), activation-based methods [9,10,22] suffer from unwanted traces for the secondary class (cat), as the model should highlight salient regions for the primary class (dog) only. To address the appearance of the secondary class due to the weighted multiplication, we first look at the problem setup. For a given image, we introduce a scheme that produces a fixed number of feature maps before going into the dense layers. This fixation limits our search space for the saliency map, and the only variables are the gradient maps that change according to the class.

Gradient matrix to a single value conversion eventually reduces many significant gradients from influencing the overall accumulation. If we keep all the gradients and perform elementwise multiplication with feature maps, the obtained map contains better representations of the assigned class. The acquired map is the intermediate global guidance map for the usual local weighted multiplication. We use this guidance map to constrain the generated feature maps to be responsive only to the assigned class through element-wise matrix multiplication. After that, we perform the formal weighted accumulation for acquiring the saliency map, followed by a carefully designed upscaling process. Thus, the produced saliency map is more responsive to the given class and its boundaries than previous approaches. The following lines summarize our overall contributions:A new saliency map generation scheme is formulated by introducing a global guidance map that incorporates the elementwise influence of the gradient tensor onto the saliency map.Acquired boundaries for the saliency maps are crisper than contemporary studies and perform efficiently within single/multi-class/multi-instance-single class cases.To validate our study, we perform seven different metric analyses on three different datasets, and the proposed study achieves state-of-the-art performance in most cases.

The rest of the paper is organized as follows: Section 2 describes the literature that is related to our method. Following this, we describe our proposed architecture with a mathematical description in Section 3. Subsequently, Section 4 contains the performance evaluation based on different relevant metrics. Section 5 highlights the advantages and limitations of the proposed method. Finally, the last section concludes with future works.

## 2. Related Work

### 2.1. Backprop-Based Methods

Zisserman et al. [1,2] first introduced gradient calculation by focusing on computing the confidence score for explanation generation, and other back propagation-based explanation studies have been introduced later [11,12,14]. However, their gradient employment and manipulation lead to several issues, as addressed by [11,12]. Instead of a single layer gradient, Srinivas et al. [11] focuses on aggregating gradients from all convolutional layers.

### 2.2. Activation-Based Methods

Class activation mapping (CAM) methods are based on the study [7], where authors select feature maps as a medium for creating an explanation map. GradCAM [9] is a weighted linear combination of the feature maps followed by the ReLU operation for a given image and the respective model. Later, GradCAM++ [10] introduced the effect of the higher-order gradients with inclusive only to positive elements properties. In this way, they achieve a more precise representation compared to the previous studies. EFC-CAM [23] has applied constraints on the weighted values of GradCAM for better performance. However, gradients are not the only way to generate the saliency map, which inspires the ScoreCAM [22], AblationCAM [13]. X-gradCAM [8], an extension to the GradCAM, follows the same underlying weighted multiplication for GradCAM. EigenCAM [24] introduces principal component analysis in developing the saliency map. CAMERAS [25] utilizes basic CAM studies with multiscale input extension, leveraging the fusion technique of multiscale feature and gradient map weighted multiplication.

However, GradCAM [9], and relevant studies have shown to produce traces of secondary saliency when images have multiple classes, as shown in Figure 1. Moreover, the boundaries of the salient regions are blurred for most results. We believe these negative effects are due to their averaged implementation of gradient maps for saliency calculation. In this paper, we introduce a global variable that element-wisely weighs the gradient map to produce clearly bounded saliency map that is free from secondary traces.

### 2.3. Perturbation-Based Methods

Another group of studies treats the neural network as a “white box” instead of a “black box” and proposes an explanation map by probing the input space. These studies generate saliency maps by checking the response to the manipulated input space. By blocking/blurring/masking some of the regions randomly, these studies observe the forward pass response for each case and finally aggregate their decision to develop the saliency map. RISE [4] first introduces such analysis, followed by external perturbation [20,21] where authors rely on the optimization procedure. SISE [5] presents feature map selection from multiple layers, followed by attribution map generation and mask scoring to generate the saliency map. Later, they improvise it through adaptive mask selection in the ADA-SISE [6] study.

## 3. Proposed Methodology

This section will describe the formulation of the proposed approach as shown in Figure 2 for obtaining the saliency for the given model and image.

### 3.1. Baseline Formulation

Every trained model infers through the collective response of its feature maps for the given image. The current understanding of deep learning requires more research to define the ideal formulation for the extracted feature maps during inference. In this subsection, we first revisit the baseline formulation used in many of the previous CAM and GradCAM-like methods [7,9,10,11,12].

For any image $X\in R^{W\times H\times 3}$; the model θ generates feature maps M during inference. Say, $M^{k_{l}}$ is the $k^{\mathrm{th}}$ feature channel at $l^{\mathrm{th}}$ layer, before the final dense layer. CAM [7] aggregates them all and obtains a unified map that will represent the collective correspondence due to the model θ for the given image. This aggregation for the activated channels is as follows:(1)$$A^{l}=\sum_{k}M^{k_{l}}$$

This tensor $A^{l}$ contains the global representation for all activated feature maps, which can serve the purpose of marking salient regions with careful tuning as shown in Figure 1. In Equation (Equation 1), we aggregated all of the feature representations into $A^{l}$; elements over a specific threshold may correspond to the primary class information. Nonetheless, this observation only works within images with a single class, but is not appropriate for the dual-class scenarios, as shown in Figure 1.

To achieve class discriminatory behavior, researchers [9,10,11,12] worked with the idea of using weighted aggregation for Equation (Equation 1), in contrast to the plain linear addition operation. To extract weights, first they compute the gradient maps $\nabla_{C}$ with respect to the given class C, from the feature maps M.

If $Y^{C}$ is the class score [9] for the given image from the input model θ, for each location (i,j) of the $k^{\mathrm{th}}$ feature map at the $l^{\mathrm{th}}$ layer, then the corresponding gradient map is expressed as: (2)$$\nabla_{C_{ij}}^{k_{l}}=\frac{\partial Y^{C}}{\partial M_{ij}^{k_{l}}}$$

If each feature map holds Z number of elements, then the corresponding weight for each feature map $M^{k_{l}}$ is:(3)$$\lambda_{C}^{k_{l}}=\frac{1}{Z}\sum_{i}\sum_{j}\nabla_{C_{ij}}^{k_{l}}$$
which is the mean value of $\nabla_{C}^{k_{l}}$. Hence, the regular baseline formulation [9] for the saliency map $S_{C}$ estimation is expressed as follows:(4)$$S_{C}=\mathrm{ReLU}\left(\sum_{k}\lambda_{C}^{k_{l}}\times M^{k_{l}}\right)$$

### 3.2. Incorporating Global Guidance

Equation (Equation 4) achieves class discriminatory behavior; however, it still faces challenges due to its formulation. If we investigate the above equation, we see that $\lambda_{C}^{k_{l}}$ weighs the corresponding feature map $M^{k_{l}}$. Typical $\lambda_{C}^{k_{l}}$ treats every member of the given $M^{k_{l}}$ equally and increases/decreases their collective effect homogeneously. Therefore, we can still see the traces of the other classes in the saliency map, as shown in Figure 1. Other studies [5,6] have shown class discriminatory performance through perturbations without relying upon the gradients. However, those studies are time expensive [11] and often require human interaction. Hence, it is natural to ask, can we still rely on the gradient-weighted operation and address the above issues?

In response to this, we propose a global guidance map. To obtain the global guidance map, we perform a simple elementwise multiplication between the feature maps and their corresponding gradient maps. The formulation for the global guidance map ($G_{M}$) is as follows:(5)$$G_{M}=\mathrm{ReLU}\left(\sum_{k}\nabla_{C}^{k_{l}}⊙M^{k_{l}}\right)$$

The idea behind considering the global guidance map is to focus only on the salient regions by limiting the operating zone for the $\lambda_{C}^{k_{l}}$. Since the multiplication operation of Equation (Equation 5) investigates individual responses for every element of the gradient maps, we can mark the class-specific regions from the feature maps. In this way, the captured guidance map successfully omits the trace of the secondary classes from Equation (Equation 4) unless the generated feature maps heavily overlap between categories, signifying the possible misclassification.

In summary, we first compute $G_{M}$ and multiply it with each of the feature maps to obtain the class discriminative feature maps from the initial class representative feature maps. Then, we perform the weighted multiplication between each member of the $\lambda_{C}^{k_{l}}$ and class discriminative feature maps, followed by the final aggregation to obtain the desired representation. Hence, our proposed weighted-multiplicative-aggregation is as follows:(6)$$S_{C}=\mathrm{ReLU}\left(\sum_{k}\lambda_{C}^{k_{l}}\times (G_{M}⊙M^{k_{l}}\right))$$

In Equation (Equation 6), the only reason we are using the $\lambda_{C}^{k_{l}}$ is to gain a similar homogeneous increment as in Equation (Equation 4), which also helps to preserve visual integrity during the final upsampling operation. Since guidance map $G_{M}$ can successfully omit the secondary classes from any $k^{\mathrm{th}}$ feature map $M^{k_{l}}$ by ‘masking’ the primary governing region, it also becomes possible to produce increments in the desired class region with the additional multiplication help from $\lambda_{C}^{k_{l}}$. Finally, typical smoothing and normalization are performed on the saliency map before post aggregation up-sampling onto the given image size. In this way, the achieved saliency map $S_{C}$ shows significant improvement in both single class representative and multiclass discriminative cases. Figure 3 illustrates each of the stages.

## 4. Performance Evaluation

**Datasets.** Our experimental setup covers three widely used vision datasets: ImageNet, MS-COCO 14, and PASCAL-VOC 12. Among them, PASCAL-VOC 12 provides the full segmentation annotations for the input images. Hence, experiments using segmentation-oriented metrics are performed on this dataset. Other experiments do not involve segmentation labels, as the rest of the experiments are applicable for all datasets mentioned above.

**Compared studies.** For visual comparison, given the availability and functional complexities, we consider available official implementations of the GradCAM [9], GradCAM++ [10], X-gradCAM [8], ScoreCAM [22], Fullgrad [11], Smooth-Fullgrad [11], CAMERAS [25], Integrated [12], and Relevance-CAM [26]. We report seven different quantitative analyses in addition to the visual demonstration. We extract saliency maps from pretrained VGG16 and ResNet50 networks for all of the methods above in our experiments. A later section reports the quantitative results from ResNet50 in the respective tables. Like previous studies [8,10,12,22], we randomly selected a few thousand random images from the ImageNet and MS-COCO 14 datasets for our experiments. However, since the random images from previous studies are not made public, the results of our random selections may vary from the cited studies.

### 4.1. Visual Demonstration

In this section, we show a qualitative comparison with previous methods. In Figure 4, we have included diverse sets of images from the datasets. The first five rows show the performance due to ResNet50, and later are for VGG16. Our image selection includes class representative, class discriminative, and multiple instances examples. Here, the proposed method can capture the class discriminative region with greater confidence, if not the entire class area itself for the bicycle, sheep, sea-bird, and dog image. Our scheme captures the class region for the sea-bird image and excludes its reflection in the water compared to other methods.

Additionally, the proposed CAM bounds the whole dog class as a dog and captures the sheep in a lowlight environment more clearly than other methods.

For images with dual classes, our method presents superior class discriminative performance. Our study successfully bounds the ’Horse’ as the primary class in the two Horse images without leaving any traces of the secondary class ’Person’. On the other hand, compared studies often mark both horse and secondary class instances. Similarly, in Mastiff image, our method can cover the utmost salient region of the primary class ’Mastiff’ without leaving any traces of the secondary class ’tiger cat’. Our concrete saliency maps without any traces show that our method can assume better model accuracy than other methods in the case of images under challenging visual conditions.

### 4.2. Quantitative Analysis

To present our quantitative analysis, we perform the following experiments: model’s performance drops and increments due to the salient and context regions, Pointing score, Dice score, and IoU score. We present the scores for the above metrics for ResNet50 for all datasets.

**Performance due to the saliency region.** If we have a perfect model and a perfect interpreter to mark the spatial correspondence for the specific class, the network will provide a similar prediction for both the given image and the segmented salient image. Here, we first extract the salient region from the given images with the help of the given interpreter. Then, we perform prediction on the original image, and the corresponding salient image [10] and check the performance drop for the given interpreter. The expectation is that the better interpreter can exclude the non-salient region as much as possible; hence, the performance drop will be as low as possible. Therefore, our first metric delivers the performance drop due to salient area only as input. For some cases, prediction performance hinders due to the presence of a strong spatial context.

**Performance due to the context region.** As above, if saliency extraction is as good as one expects, then we can set up another experiment with the context region. In this setup, we first exclude the salient part from the given image to obtain the context image and predict it. If the interpreter can successfully extract all the salient areas, the performance will drop near 100 percent.

**Pointing game, Dice Score, IoU.** For image sets with segmentation labels, various segmentation evaluations can be calculated for saliency maps. We follow [17] to perform the pointing game for class discriminative evaluation. In this performance metric, the ground-truth label is used to trigger each visualization approach, and the maximum active spot on the resulting heatmap is extracted. After that, it determines if the highest saliency points undergo into the annotated boundary box of an object, determining whether it is a hit or miss. The pointing game accuracy can be expressed as follows:(7)$$\mathrm{pointing}\;\mathrm{game}=\frac{Hits_{total}}{Hits_{total}+Misses_{total}}$$

The high-pointing game value represents a better explanation for the model. The dice score is a popular metric for analyzing segmentation performance. It results from the ratio of the doubled intersection over the total number of elements associated with this instance. IoU stands for intersection over the union. It is a widespread metric for evaluating segmentation performance. This score ranges from 0 to 1 and signifies the overlapping area between the obtained image and its corresponding ground truth.

The proposed method performs better or similar to the best-performing saliency generation method in Table 1. Here, we present the comparative data on the Pascal VOC 2012 dataset for the ResNet50 model. Out of seven different performance tests, our method obtains the highest score for six of them. For an increase in the context zone, our study differs only 0.03 from the best performing result. In Table 2, the proposed research avails state-of-the-art performance for three out of four metrics. Table 3 also shows the best performance of our method for every experiment metric using the ImageNet dataset. The PASCAL VOC 2012 dataset has a corresponding segmentation mask, and we can achieve pseudo segmentation masks by thresholding the saliency maps. Achieved scores for the Pointing game, Dice and IoU signify that our study can better capture interest zones than the compared studies. To measure the explainability, we also conducted a comparative analysis in Figure 5 on three images from the Pascal VOC 2012 dataset and presented the insertion and deletion metrics proposed in [4]. Here, our method captures the most salient regions for both single, dual, and dual classes with multiple instances in comparison to GradCAM [9], GradCAM++ [10], and X-gradCAM [8].

### 4.3. Interpretation Comparison

Saliency map generation is not all about capturing the class of interest as precisely as possible. A faithful interpretation is also a significant part of saliency generation studies. In other words, any interpretable method should explain why the underlying model is making such a prediction by marking the corresponding image region. However, we cannot present this for every image from the dataset, but a sophisticated example can show the difference from previous methods.

In Figure 6a, we have presented the interpretation comparison between the proposed study and the GradCAM++ [10]. Here, the top-1 class response is ‘Albatross’ for the given image. For VGG16, the saliency map for the proposed method marks one of the Albatross birds and the water as context, but fails to mark the other Albatross bird. For ‘Albatross’, our method with the VGG16 shows different interpretations due to the global guidance map and responds to the water as context. In contrast to ResNet50, our guidance map marks both Albatross birds without marking the water context. However, for both networks, GradCAM++ [10] captures both of the birds and also barely touches the water context. For this particular image, the proposed method presents a clearer interpretation difference between VGG16 and ResNet50.

In Figure 6b, we show why the models might identify the given image as ‘Ruddy Turnstone’ class. With our scheme, we interpret that surrounding stones and water are features that are corresponding to the Ruddy Turnstone bird for both VGG16 and ResNet50. This interpretation makes more sense if we look at typical Ruddy Turnstone images from the ImageNet dataset Figure 6b, where the most of Ruddy Turnstone birds are shown with a stony sea-shore area as the background. Hence, we can utilize this interpretation as a medium for identifying the dataset bias. On the other hand, GradCAM++ [10] shows the Albatross bird regions as the interpretation for the Ruddy Turnstone, and almost ignores the associated context.

## 5. Discussion

Several studies [22,27] have raised some questions about the faithfulness of gradients to explain deep networks. The deeper depth of the model may increase the noise and discontinuity of the gradients, which is called the shattered gradient problem [27]. Passing through the activation functions such as ReLU or Sigmoid may result in saturation or explosion of gradients, and gradients can get noisy due to this saturation. GradCAM [9] and GradCAM++ [10] accumulate only positive gradients for weighing the features, which makes them more susceptible to these noises. Our method incorporates all of the gradients to produce our global guidance map that generates the salience map with local information of the corresponding classes. As a result, our method can produce a crisper saliency map without traces of unwanted classes. The visual saliency maps from explainable methods can be used for segmentation and localization under complex supervision. Our crisper saliency map can achieve significant performance upgrades in both single-class and multi-class segmentation and localization applications.

Different class-discriminative methods backpropagate different values such as gradients for GradCAM [9,10], relevance-score [26] or maximum activated regions [28]. As an extension of [9], our method also backpropagates gradients that require an additional computational burden. Moreover, also like others, our method is feature map-dependent. Comparatively shallower networks might produce faulty feature maps which are responsible for faulty gradients under the corresponding class. These faulty gradients can create “false positive regions” that result in poor segmentation performance over the target class and poorer results for segmentation-based metrics such as Dice and IoU. Finally, distinguishing multiple instances of the same class is still an unapproachable problem using gradient backpropagate-based scheme.

## 6. Conclusions

In this study, we present a novel extension of the traditional gradient-dependent saliency map generation scheme. The proposed method leverages element-wise multiplicative aggregation as guidance with previous weighted multiplicative summation and further improves the performance of salient region bounding. Additionally, we showed our study’s advanced class discriminative performance and presented evidence for better area framing with deeper networks. Furthermore, our model can avail a crisper saliency map and significant quantitative improvement over three widely used datasets. High-level vision tasks such as segmentation, object detection and localization require highly class-discriminative behavior of the model for better performance. We aim to apply our crisper saliency map for segmentation using only the classification data. We are also interested in the problem of distinguishing different instances of the same class.

## Figures and Tables

**Figure 1 sensors-22-06516-f001:**
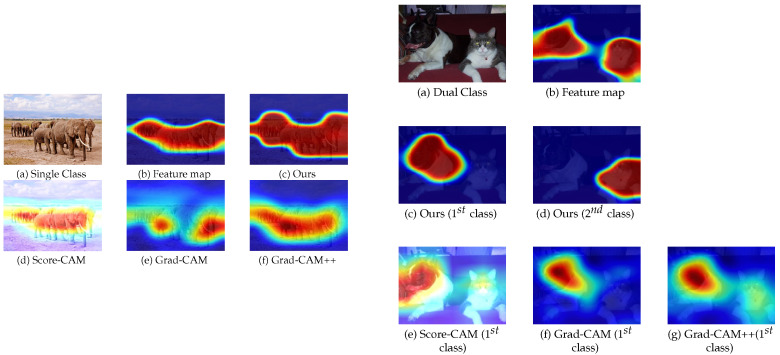
Examples of feature maps, saliency maps from three existing methods [9,10,22], and proposed method’s results for single (**left**) and dual-class (**right**) images. For the single class, our method’s response is close to the feature map response. For dual-class, our method shows the class discriminatory behavior, whereas the other methods still show the traces of the secondary class on the primary class response.

**Figure 2 sensors-22-06516-f002:**
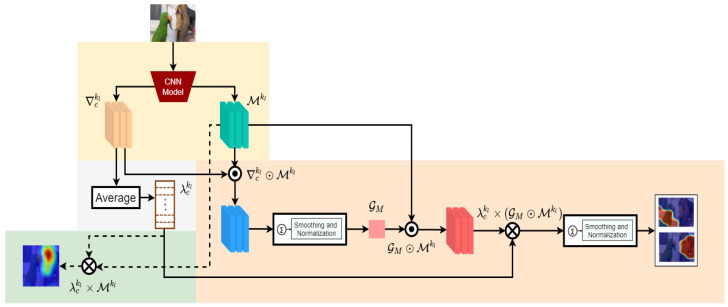
A comparative overview of the proposed saliency generation. In the baseline formulation [9], the $\lambda_{C}^{k_{l}}$ (the mean value of $\nabla_{C}^{k_{l}}$) is directly multiplied to the feature map $M^{k_{l}}$ to produce saliency map, Equation (Equation 4) (left green). In our scheme, first we build the global guidance map $G_{M}$ from gradients and feature map, Equation (Equation 5). Then, we follow the baseline formulation by multiplying $\lambda_{C}^{k_{l}}$ and guided feature maps to produce the refined saliency map.

**Figure 3 sensors-22-06516-f003:**
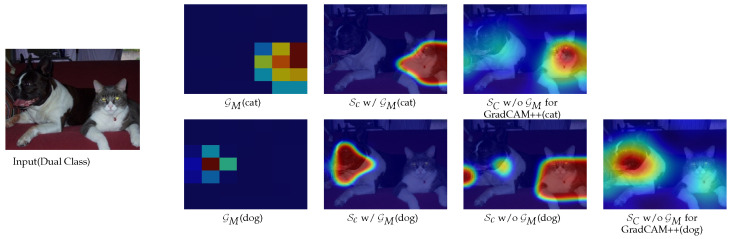
Proposed global guidance maps $G_{M}$ for cat, proposed saliency map $\left(S_{c}\right)$ with $G_{M}$ for cat and saliency map $\left(S_{c}\right)$ of Grad-CAM ++ [10] without global guidance $G_{M}$ (first row), and the same maps for the dog [second row]. The proposed global guidance map provides strong localization information as well as the exclusion of non-target class.

**Figure 4 sensors-22-06516-f004:**
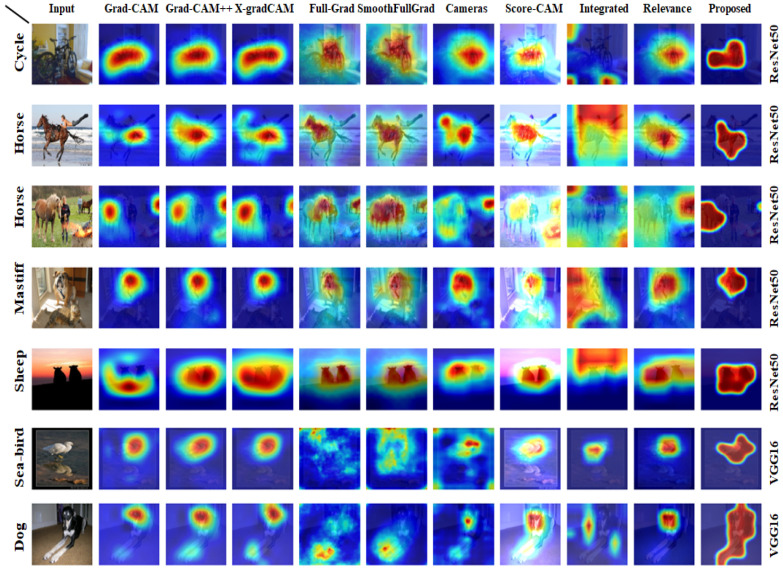
Visual comparison between previous state-of-the-art studies and the proposed method. For the given demonstrations, our approach can mark down the primary salient regions under challenging visual conditions. Additionally, our saliency maps are more concrete and leave almost no traces for the secondary-salient areas.

**Figure 5 sensors-22-06516-f005:**
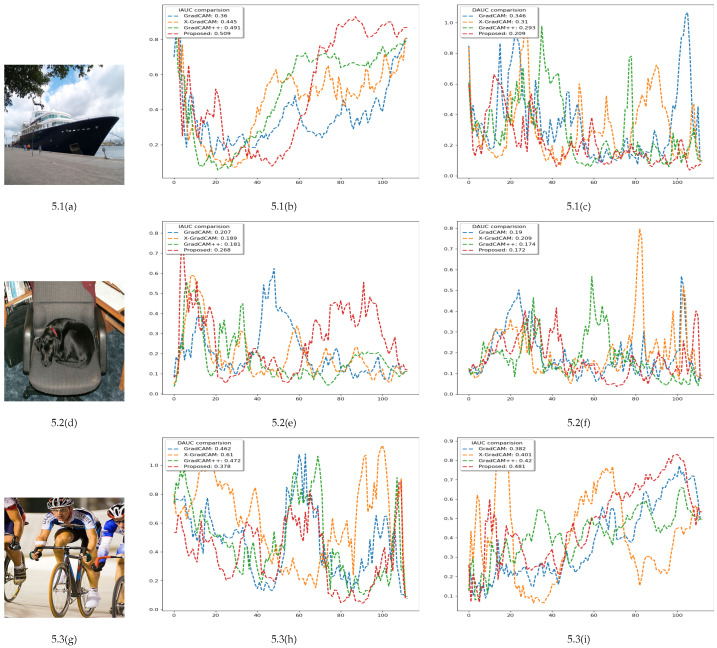
AUC (Area under curve) demonstration for the insertion and deletion operation for images on the left side. The above analysis shows that the proposed method can capture the most salient regions for a single class, dual-class, and dual-class with multiple instances compared to the previous methods [8,9,10]. The second column demonstrates the IAUC (Insertion area under curve) comparison and the third column demonstrates the DAUC (Deletion area under curve) comparison.

**Figure 6 sensors-22-06516-f006:**
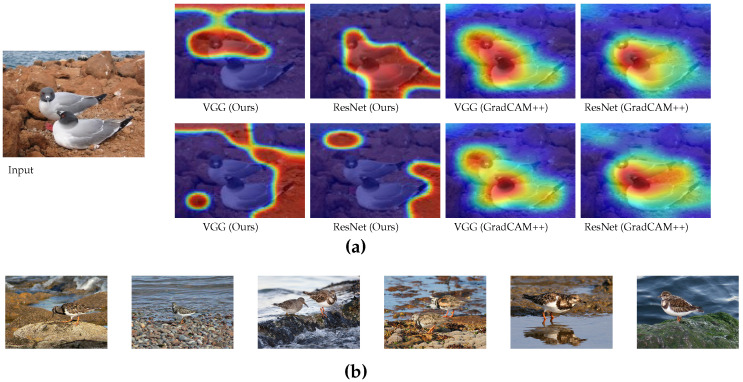
Interpretation comparison between the proposed and GradCAM++ [10] to explain why the underlying model is making such a prediction by marking the corresponding image region. (**a**) Interpretation comparison between the proposed and GradCAM++ [10]. The upper row is the response map for the primary ’Albatross’ bird class, and the second row is for the secondary class, ’Ruddy Turnstone’ bird. The proposed method clearly presents the difference between VGG16 and ResNet50 in terms of interpretation, whereas GradCAM++ [10] responses are all similar across different networks and classes. (**b**) Upon examining typical Ruddy Turnstone bird images in the ImageNet dataset, we see that stony shore is the background for most of the Ruddy Turnstone bird images and we can see how stony shore is interpreted as ’Ruddy Turnstone’ for VGG Net.

**Table 1 sensors-22-06516-t001:** Comparative evaluation in terms of salience zone drop and context zone increase, Pointing Game, Dice, IoU (higher is better) and salience zone increase and context zone drop (lower is better) on the PASCAL VOC 2012 dataset for ResNet50 model. Comparison performed on GradCAM, GradCAM++, X-gradCAM, CAMERAS, FullGradCAM, Smooth FullGradCAM, Integrated GradCAM, ScoreCAM, RelevanceCAM, and our proposed method, respectively. The best scores are in bold form and second-best scores are underlined.

	[9]	[10]	[8]	[25]	[11]	[11]	[12]	[22]	[26]	Proposed
Increase for salience zone ↑	0.0366	0.0716	0.0366	0.0715	0.0421	0.0421	0.0172	0.0506	0.0559	**0.0786**
Drop for context zone ↑	0.9395	0.8812	0.9429	0.8834	0.9083	0.9281	0.8124	0.9389	0.9089	**0.9443**
Pointing Game ↑	0.3355	0.4731	0.3733	0.4412	0.3713	0.4422	0.2642	0.4322	0.532	**0.5945**
Dice ↑	0.2822	0.3422	0.2934	0.3342	0.2942	0.3328	0.1834	0.3321	0.411	**0.4321**
IoU ↑	0.0823	0.1122	0.0901	0.1007	0.0812	0.0963	0.0625	0.0943	0.121	**0.1321**
Drop for salience zone ↓	0.8996	0.8064	0.8745	0.8201	0.8873	0.8762	0.9399	0.8567	0.8333	**0.7784**
Increase for context zone ↓	0.0172	0.0312	0.0205	0.0244	0.0291	0.0215	0.0411	**0.0151**	0.029	0.0183

**Table 2 sensors-22-06516-t002:** Comparative performance drop and increment of saliency and context zones on the MS-COCO 14 dataset for ResNet50 model. Comparison performed on GradCAM, GradCAM++, X-gradCAM, CAMERAS, FullGradCAM, Smooth FullGradCAM, Integrated GradCAM, ScoreCAM, RelevanceCAM, and our proposed method, respectively. The best scores are in bold form and second-best scores are underlined.

	[9]	[10]	[8]	[25]	[11]	[11]	[12]	[22]	[26]	Proposed
Drop for context zone ↑	0.9023	**0.9495**	0.9142	0.9424	0.8961	0.9025	0.8607	0.9183	0.9081	0.9391
Increase for saliency zone ↑	0.0490	0.0913	0.0555	0.0935	0.0455	0.0495	0.0796	0.0695	0.089	**0.0999**
Drop for saliency zone ↓	0.8394	0.7243	0.8081	0.7325	0.8454	0.8288	0.7713	0.7822	0.7357	**0.6493**
Increase for context zone ↓	0.0245	0.0165	0.0215	0.0172	0.0311	0.0315	0.0335	0.0205	0.0311	**0.0152**

**Table 3 sensors-22-06516-t003:** Comparative performance drop and increment of saliency and context zones on the ImageNet dataset for ResNet50 model. Comparison performed on GradCAM, GradCAM++, X-gradCAM, CAMERAS, FullGradCAM, Smooth FullGradCAM, Integrated GradCAM, ScoreCAM, RelevanceCAM, and our proposed method, respectively. The best scores are in bold form and second-best scores are underlined.

	[9]	[10]	[8]	[25]	[11]	[11]	[12]	[22]	[26]	Proposed
Drop for context zone ↑	0.8767	0.9178	0.8698	0.9335	0.8379	0.8548	0.8938	0.8585	0.8541	**0.9392**
Increase for saliency zone ↑	0.0535	0.0903	0.0635	0.0936	0.0803	0.0669	0.0435	0.1003	0.0969	**0.1008**
Drop for saliency zone ↓	0.7906	0.6717	0.7682	0.7302	0.7775	0.7844	0.7267	0.6975	0.6844	**0.6492**
Increase for context zone ↓	0.0234	0.0067	0.0368	0.0134	0.0502	0.0301	0.0301	0.0468	0.0602	**0.0012**

## Data Availability

The data that support the findings of this study are openly available in ImageNet at https://www.image-net.org/, Pascal-VOC at http://host.robots.ox.ac.uk/pascal/VOC/, and MS-COCO at https://cocodataset.org/.

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
