# Peer review of "Rethinking Gradient Weight’s Influence over Saliency Map Estimation"

_sensors, 2022, doi:10.3390/s22176516_

Round 1
Reviewer 1 Report
In this paper, the authors introduce a way to rectify the weighted aggregation operation during saliency estimation called global guidance map. This last is obtained through elementwise multiplication between the feature maps and their corresponding gradient maps. The authors compare their proposal with nine different saliency approaches on the ImageNet, MS-COCO 14, and PASCAL VOC 2012 datasets. The results suggest that the global guidance map improves the quality of saliency maps. However, in my opinion there are still some issues that jeopardize the contribution of this paper:
- There are still some typos (e.g. g_M instead of G_M in the caption of Figure 3, "Integrated[12]" is out of the layout, "due the to"). I suggest careful proofreading.
- The sentence "still contains traces of unwanted classes" is understandable but I suggest adding a few more words to explain your point.
- In the Related Literature, I suggest highlighting the differences between your approach and the cited ones in order to strengthen your proposal.
- "Z number of elements" is too broad. Does "elements" stand for pixels, matrix values, or something else?
- In Section 3.1, much of the theory seems to come from Grad-CAM. If that's the case, I suggest stating clearly at the beginning of the section that this theory belongs to Grad-CAM and it is the starting point of your approach.
- In Section 3.2, global guidance map G_M seems referring to all the classes present in the dataset, while in the formula we have the gradient^k_l of a specific class. Should G_M be named G_{M_C}?
- It is not clear why in Equation 6, there is the elementwise multiplication between G_M and M^{k_l} even if M^{k_l} is already present in the computation of G_M. Could the approach work without that multiplication?
- In Sections 4.1 and 4.2, there is some confusion about quantitative and qualitative results. Section 4.1 lasts only 4 lines and does not explain the quantitative results of the experiments. Section 4.2, which is the one devoted to qualitative results, contains paragraphs such as "Pointing game, Dice Score, IoU" that explain the quantitative results present in the tables. I suggest organizing the qualitative and quantitative analysis in a different (and clear) way.
- Before Conclusion, it could be interesting to add a Discussion section, in which you wrap up the advantages of the proposal and its limitations (such as when you cannot use it, if it works better with fewer classes, etc.).
- There is no plan of the paper at the end of the Introduction.
- I suggest adding more sentences to explain the next steps and adding two/three ideas of possible future works.
Author Response
We thank the reviewers for detail comments and questions.
Please see the Report Notes for all the point-by-point responses.
Ho Yub jung

Reviewer 2 Report
The feature maps and accompanying gradient maps were elementwise multiplied by the authors to produce the global guidance map. Nine different saliency visualizers were put to the test against the proposed study.
Comments
All mathematical equations must remark by its references, otherwise the authors must give a complete derivation. For example, Eq.(1) is done by the authors or brought it from elsewhere ?
Secondly what is the notion≺? In my field in analysis this means the subordination, in algebra means a relation,… etc . All these notations should be clear for the readers and makes the paper readable in all the times.
The problem statement is not clearly described. The authors need to add more elaboration and motivation.
I think there is a mistake with the comparison tables, where there are duplicate references [11];the authors must correct it in all Tables 1,….
The term in Line 222, must suppurate in new line with a given reference;
More discussion on Fig.4 is need it;
I recommend it for publication.
Author Response

(The authors gave the same response as above.)

Round 2
Reviewer 1 Report
The authors addressed all my concerns. I think that this paper is ready for publication.
Reviewer 2 Report
Accepted.